# Supporting High-Stakes Decision Making Through Interactive Preference Elicitation in the Latent Space

**Michael Eichelbeck,**[*] **Tim Voigt,**[*] **Matthias Althoff**
Department of Computer Engineering
Technical University of Munich, Germany
`{michael.eichelbeck, tim.voigt, althoff}@tum.de`

## Abstract

High-stakes, infrequent consumer decisions, such as housing selection, challenge conventional recommender systems due to sparse interaction, heterogeneous multi-criteria objectives, and high-dimensional features. This work presents an interactive preference elicitation framework utilizing preferential Bayesian optimization (PBO) to learn the unknown utility function of a user from pairwise comparisons that are integrated in real-time. To increase efficiency in a complex feature space, we learn the preference model in the latent space of an autoencoder (AE). Additionally, to mitigate a cold start, we obtain a personalized probabilistic prior through an automated user interview with a large language model (LLM). We evaluate the developed method on rental real estate datasets from two major European cities. The results show that executing PBO in the AE latent space improves final pairwise ranking accuracy by 12%. For LLM-based preference prior generation, we find that direct, LLM-driven weight specification is outperformed by a static prior, while probabilistically weighted priors using LLMs achieve 25% better pairwise accuracy.

## 1 Introduction

User-tailored recommendations are a cornerstone of modern markets and online platforms, aiming to present the most relevant options to reduce decision paralysis and increase click-through rates. Traditional recommendation approaches excel in entertainment or e-commerce domains, where user behavior generates abundant implicit feedback through clicks, purchases, and ratings. However, they struggle in sparse-feedback environments, where users interact with only a handful of options before committing to one, motivating strategies for eliciting preferences in an interactive manner with as few interactions as possible. Such environments often correspond to high-stakes infrequent decisions characterized by complex heterogeneous multi-dimensional preference spaces; they have received little attention in the recommender system literature. We focus on the real estate market as an exemplary case study of an underexplored domain (Gharahighehi et al., 2021).

### 1.1 Related Work

**Classical Preference Elicitation**  Traditional preference elicitation methods include conjoint analysis for estimating utilities over multi-attribute items (Arora & Huber, 2001) and multi-armed bandit approaches that balance exploration and exploitation (Parapar & Radlinski, 2021). Recent multi-armed bandit extensions incorporate knowledge graphs to model inter-item relations and improve elicitation efficiency (Zhao et al., 2022). Preferential Bayesian optimization (PBO) adapts Bayesian optimization (BO) principles to scenarios lacking explicit objective functions by using implicit feedback like pairwise comparisons (González et al., 2017). Subsequent work has focused on developing acquisition functions accounting for uncertainty in both model predictions and user responses (Astudillo & Frazier, 2020; Astudillo et al., 2023).

---

[*]Equal contribution

**Preference Elicitation with LLMs**    Preference elicitation based on large language models (LLMs) follows two main approaches. Conversational methods enable dynamic natural language dialogues, using methods like GATE to actively elicit user intent through open-ended interactions (Li et al., 2025; Andukuri et al., 2024). Structured approaches integrate LLMs within probabilistic frameworks, combining language models with Bayesian methods. One type of structured elicitation uses LLMs for user interaction and Bayesian methods for maintaining preference beliefs (Handa et al., 2024; Austin et al., 2024). Similar approaches fine-tune LLMs in a supervised manner using probabilistic models, yielding improved conversational preference elicitators (Piriyakulkij et al., 2023; Qiu et al., 2026). Here, Bayesian methods are only used for fine-tuning the model and do not directly aid in question selection and recommendation. Related work explores LLMs for decision support by constructing utility functions from stated user goals (Liu et al., 2025), which does not incorporate Bayesian methods but relies on Monte Carlo simulations for expected utility maximization. Existing work has focused on discrete feature spaces, such as category labels. For feature spaces with several continuous dimensions, natural language representations are inefficient. To still leverage LLMs, we propose open-ended conversations solely to generate personalized priors for downstream PBO tasks.

**Bayesian Optimization in High-Dimensional Spaces**    Real-world recommendation scenarios frequently involve high-dimensional feature spaces that challenge conventional PBO approaches, as the search space grows exponentially with each additional dimension (Bellman, 1966). Two strategies address the curse of dimensionality in BO. The first explores lower-dimensional subspaces iteratively via one-dimensional subspace exploration for high-dimensional PBOs (Tucker et al., 2020; Cheng et al., 2020). The second strategy performs optimization in learned low-dimensional latent spaces. One approach uses a random projection matrix to embed the doubled dueling solution space into a lower-dimensional preferential intrinsic subspace, where optimization is performed while pairwise comparisons remain in the original space, approximately preserving the preference function (Zhang et al., 2023). Another approach employs manifold Gaussian processes (GPs), i.e., GPs with a neural network embedded in the covariance function, to jointly learn a nonlinear feature mapping, the response surface, and a multi-output GP reconstruction mapping, enabling acquisition function optimization in the low-dimensional feature space subject to a Lipschitz continuity constraint that encourages exploration near observed data (Moriconi et al., 2020). Similarly, variational autoencoders (AEs) have been used to learn low-dimensional latent spaces for BO, for instance in constrained molecular design, where the learned representation avoids invalid regions of the search space (Griffiths & Hernández-Lobato, 2020). The ability of AEs to capture nonlinear dependencies through flexible compression makes them particularly suited for high-dimensional feature spaces with strong interdependencies. However, to the best of our knowledge, AEs have not yet been explored for interactive preference elicitation.

## 1.2 CONTRIBUTIONS

This work addresses the challenge of efficiently learning user preferences in high-dimensional, complex recommendation domains where direct preference specification is difficult, interaction data is sparse, and new data becomes available over time. We propose a comprehensive framework that couples PBO with user-specific warm-start prior elicitation through LLMs, and AE-based feature embeddings. This facilitates preference learning in a low-dimensional latent space.

We evaluate our approach in the context of recommendations for renting real estate. While this serves as an example for a challenging high-stakes domain, our approach generalizes to other domains with similar characteristics, such as the automotive or financial services markets. Through LLM-based and statistics-based user simulations, we demonstrate that our framework is real-time capable and outperforms vanilla PBO on two real-world datasets of the real estate markets in Madrid, Spain, and Munich, Germany.

The remainder of this paper is organized as follows. After introducing some preliminaries (Sec. 2), we pose our problem statement (Sec. 3.1) and detail the proposed framework (Sec. 3.2). This is followed by the evaluation of our case study (Sec. 4) and the conclusion (Sec. 5).

## 2 PRELIMINARIES

**Preference Learning**   Preference learning is concerned with learning models from empirical preference data. A preference can be conceptualized as a "*relaxed constraint which, if necessary, can be violated to some degree*" (Fürnkranz & Hüllermeier, 2011). Common approaches range from approximating individual utility functions to applying collaborative filtering across diverse user populations. Preference learning constitutes two primary problem types: learning utility functions and learning preference relations (Fürnkranz & Hüllermeier, 2011). A typical task involves learning a function that predicts preferences for an unseen set of items, based on a known set of preferences. This work focuses on the object ranking task. The objective is to learn a function that produces a total ordering of a set of objects without access to explicit class labels – a form of unsupervised learning.

**Bayesian Optimization**   BO provides a sample-efficient framework for global optimization of expensive, black-box functions. It fits a probabilistic surrogate to the unknown objective and uses an acquisition function to decide what to evaluate next, balancing exploration and exploitation in a principled way (Frazier, 2018). In this work, BO maximizes an unknown function $f : \mathcal{X} \to \mathbb{R}$ over a compact feature space $\mathcal{X} \subset \mathbb{R}^d$. At iteration $k$, we observe noisy evaluations

$$y_k = f(x_k) + \varepsilon_k, \quad \varepsilon_k \sim \mathcal{N}(0, \sigma^2),$$

and collect observations $\{(x_i, y_i)\}_{i=1}^k$. A common surrogate for the black-box function is a GP prior, which yields a Gaussian posterior at any $x$ conditional on the observations collected so far. The subsequent evaluation maximizes an acquisition function $\alpha_k(x)$ that quantifies the value of sampling at $x$. These acquisitions should be cheap to evaluate, and several options have been proposed in the literature (Brochu et al., 2010; Astudillo et al., 2023; González et al., 2017). BO alternates between updating the surrogate with the new observation, maximizing $\alpha_k(x)$ to choose $x_{k+1}$, evaluating $y_{k+1}$, and augmenting the data. It terminates upon budget exhaustion or convergence, e.g., vanishing expected improvement.

**Preference Bayesian Optimization**   PBO has the same objective as BO but operates under the assumption that direct querying of $f$ is infeasible, so we have to rely on pairwise comparisons with two objects $(x_a, x_b)$, so-called duels. In each duel, we receive binary feedback, indicating which object was selected. This dueling process is repeated until the uncertainty is reduced to a satisfying amount. Utilizing BO techniques reduces the number of duels needed, and utilizing a trained PBO model enables ranking of previously unseen items (González et al., 2017).

**Autoencoders**   AEs are neural networks that learn compact latent representations by training an encoder $g_\theta : \mathbb{R}^d \to \mathbb{R}^r$ and a decoder $h_\theta : \mathbb{R}^r \to \mathbb{R}^d$ to reconstruct inputs, where $r \ll d$ is the so-called latent dimension (Hinton & Salakhutdinov, 2006). Training minimizes a reconstruction loss, such as mean squared error for continuous features or binary cross-entropy for binary features, often with regularization (e.g., weight decay). A well-trained encoder ideally removes correlated features, captures nonlinear relationships, and distills the input into its most relevant components.

## 3 INTERACTIVE PREFERENCE ELICITATION FRAMEWORK

In this section, we present our problem statement and describe our approach in detail.

### 3.1 PROBLEM STATEMENT

We define $u \colon \mathcal{X} \to \mathbb{R}$ as the unknown utility function of a user, defined on the feature space $\mathcal{X} \subset \mathbb{R}^d$. A corresponding pairwise preference function

$$F_u(x, x') = \begin{cases} 1 & \text{if} \quad u(x) \geq u(x'), \\ 0 & \text{otherwise} \end{cases} \tag{1}$$

maps any pair of data points $(x, x')$ to a binary response, indicating which option is preferable. Our goal is to obtain a utility surrogate $\hat{u}^*$ with a corresponding pairwise preference probability

distribution $F_{\hat{u}}$, such that

$$\hat{u}^* = \arg\min_{\hat{u}} \mathbb{E}_{(x,x')\sim\text{Uniform}(\mathcal{X}^2)}\Big[L\big(F_u(x,x'), F_{\hat{u}}(x,x')\big)\Big], \tag{2}$$

where $(x, x')$ is a pair drawn uniformly at random from $\mathcal{X}^2$ and $L$ is an appropriate loss function. While a set of items $\mathcal{I} = \{x_1, \ldots, x_{|\mathcal{I}|}\}$ is known at the time of preference elicitation, new options might be unveiled over time. We assume a channel through which we can query the user by proposing pairwise comparisons and obtaining binary feedback. In a realistic setting, the number of queries is limited by an unknown budget $N \in \mathbb{N}$. Therefore, we aim to model the utility function of the user as accurately as possible with the fewest queries possible.

## 3.2 PREFERENCE BAYESIAN OPTIMIZATION IN THE LATENT SPACE

Our approach leverages AEs by performing BO in the latent space of the AE. Optimization in this reduced space should converge more rapidly while still capturing user preferences accurately. In summary, we learn a utility surrogate $\hat{u}: \mathcal{Z} \to \mathbb{R}$ in the latent space $\mathcal{Z} \subseteq \mathbb{R}^r$ of the AE, with the corresponding pairwise preference function

$$F_{\hat{u}}(x, x') = \begin{cases} 1 & \text{if} \quad \hat{u}\big(g_\theta(x)\big) \geq \hat{u}\big(g_\theta(x')\big), \\ 0 & \text{otherwise,} \end{cases} \tag{3}$$

where $g_\theta$ is the encoder of the AE trained on the set of available items $\mathcal{I}$. The associated encoder is denoted as $h_\theta$. In addition to the following textual description, our approach is formalized in Algorithm 1.

---

**Algorithm 1** Preferential Bayesian Optimization in the Latent Space

---

**Require:** Item dataset $\mathcal{I} = \{x_1, \ldots, x_{|\mathcal{I}|}\}$, where $x_i \in \mathcal{X} \subseteq \mathbb{R}^d$
**Require:** Encoder $g_\theta : \mathcal{X} \to \mathcal{Z}$, decoder $h_\theta : \mathcal{Z} \to \mathcal{X}$, where $\mathcal{Z} \subseteq \mathbb{R}^r$ and $r \ll d$
**Require:** Initialization budget $M \in \mathbb{N}$, query budget $N \in \mathbb{N}$
**Ensure:** Learned utility surrogate $\hat{u} : \mathcal{Z} \to \mathbb{R}$
    **Elicit user-specific feature weights and bounds:**
    $(\pi, \underline{x}, \overline{x}) \leftarrow \text{runLLMConversation}()$
    $s^2 \leftarrow \text{calcFeatureVariances}(\mathcal{I})$
    $w \leftarrow \text{sampleWeightsFromRanking}(\pi, s^2)$
    **Initialize model:**                                                                   ▷ 3.2.1
    $\mathcal{D} \leftarrow \emptyset$                                    ▷ Set of observations based on pairwise comparisons
    $u_{\text{lin}}(x) \leftarrow w^\top x$                                ▷ Synthetic linear utility in feature space
    **for** $k \in \{1, \ldots, M\}$ **do**
        Sample random pair $(x_k, x'_k)$ from $\mathcal{I}$
        **if** $u_{\text{lin}}(x_k) > u_{\text{lin}}(x'_k)$ **then**
            $y_k \leftarrow 1$
        **else**
            $y_k \leftarrow 0$
        **end if**
        $z_k \leftarrow g_\theta(x_k), z'_k \leftarrow g_\theta(x'_k)$                       ▷ Encode from original to latent space
        $\mathcal{D} \leftarrow \mathcal{D} \cup \{(z_k, z'_k, y_k)\}$
    **end for**
    $\hat{u}_M = \texttt{Fit}(\hat{u}_0, \mathcal{D})$                           ▷ Fit initial GP model with uninformative prior $\hat{u}_0$
    **Interactive elicitation:**                                                       ▷ 3.2.2
    **for** $k \in \{M+1, \ldots, M+N\}$ **do**
        **Active candidate selection:**
        $(z_k, z'_k) \leftarrow \arg\max_{z,z'\in[g_\theta(\underline{x}),g_\theta(\overline{x})]} \texttt{qEUBO}_k(z, z')$
        **Query user:**
        $(x_k, x'_k) \leftarrow (h_\theta(z_k), h_\theta(z'_k))$                       ▷ Decode from latent to feature space
        $y_k \leftarrow \text{getUserResponse}(\hat{x}_k, \hat{x}'_k)$
        **Update model:**
        $\hat{u}_k = \texttt{Fit}(\hat{u}_{k-1}, \{(z_k, z'_k, y_k)\})$
    **end for**
    **return** $\hat{u}_{M+N}$

---

### 3.2.1 UTILITY PRIOR ESTIMATION USING LLMS

In PBO, selecting informative duels is particularly important during the early stages of elicitation (Handa et al., 2024; Brochu et al., 2010), and an unsuitable starting point could waste valuable query budget. To mitigate this cold start, we aim to find a maximally informative prior to initialize the preference model. This is achieved by evaluating $M$ pairwise preference decisions based on a linear utility function $u_{\text{lin}}(x) = w^\top x$, where the prior weights $w \in [-1,1]^d, \|w\|_1 = 1$ are obtained through an LLM-guided user interview instead of relying on a predefined static weight vector. Our approach builds on the initialization phase of (Handa et al., 2024), extended here to handle continuous features.

**User Interview** The LLM is assigned the persona of a domain-specific interviewer. Apart from reaching the query budget, the conversation can also conclude when the LLM determines it has gathered sufficient information or when the user wishes to stop the interview. The obtained preference information $\pi$ either directly contains the utility surrogate weights $w = \pi$ or a ranking for the probabilistic initialization explained below. Additionally, lower and upper bounds $\underline{x}, \overline{x}$ of the feature subspace that is acceptable for the user are returned by the LLM. Including hard constraints can make the elicitation process significantly more efficient by ensuring that all presented comparisons fall within the acceptable feature subspace of the user. The used prompts are provided in the Appendix (Sec. A.2.2). An example output of the LLM for the real estate domain is:

1. **Lower bounds** on essential criteria, including the minimum floor level, required living area in square meters, and available parking space.
2. **Upper bounds** for constraining criteria, such as maximum acceptable total monthly rent and maximum acceptable travel time to the workplace.
3. **Feature importance weights** representing the relative significance of each feature in the decision-making process. The LLM is prompted to estimate these weights based on the conversation.

**Probabilistic Weight Initialization** Instead of directly returning utility function weights, we employ an approach based on the work in (Handa et al., 2024), which asks the LLM to rank features in order of importance – a task that aligns better with the strengths of LLMs in comparative reasoning and ordinal relationships (Liusie et al., 2024; Liu et al., 2024). This approach works by sampling feature weights from normal distributions whose parameters are informed by both the ranking of the LLM and the empirical feature variances. For each feature $i$ with rank $r_i$ (where lower ranks indicate higher importance), the weight $w_i$ is sampled from:

$$w_i \sim \mathcal{N}\left(0, \frac{s_i^2}{\max_{j \in \{0,\ldots,d\}}(s_j^2)} \cdot \frac{1}{r_i}\right), \tag{4}$$

where $s_i^2$ represents the empirical variance of feature $i$ before normalization. The intuition behind this approach is that features deemed more important by the user (receiving lower rank values) should have larger absolute weight values, while features with higher empirical variance already exhibit significant influence on the feature space and thus warrant proportionally scaled weights. Different from (Handa et al., 2024), we add $\frac{1}{\max_{j \in \{0,\ldots,d\}}(s_j^2)}$ as a normalization term ensuring that features with exceptionally large variances do not receive disproportionately large weights regardless of their actual importance to the user. The corresponding prompt for our case study is provided in the Appendix (Sec. A.2.3).

**Warm-Start Dataset** After the weights for the linear model $u_{\text{lin}}(x)$ have been determined, we sample $M$ item pairs from $\mathcal{I}$ uniformly at random. For each pair $(x_k, x'_k)$, we evaluate the associated pairwise preference function to obtain the binary feedback $y_k = F_{u_{\text{lin}}}(x_k, x'_k)$. Since the preference model is trained in the latent space, we compute the dataset of embedded observations

$$\mathcal{D} = \left\{ \left( g_\theta(x_k), g_\theta(x'_k), y_k \right) \right\}_{k=0}^{M} \tag{5}$$

as well as approximate embedded lower and upper bounds of the feasible feature subspace $\underline{z} = g_\theta(\underline{x}), \overline{z} = g_\theta(\overline{x})$. Note that since the encoder $g_\theta$ is not necessarily monotone, these bounds represent an approximation of the feasible region in latent space. In practice, we observe that this approximation works well for the learned encoders in our experiments.

### 3.2.2 Elicitation Loop

Denoting the $k^{\text{th}}$ update of the utility surrogate $\hat{u}$ based on new observation data $\mathcal{B}$ as $\hat{u}_k = \texttt{Fit}(\hat{u}_{k-1}, \mathcal{B})$, we initialize $\hat{u}$ using the warm-start dataset as $\hat{u}_M = \texttt{Fit}(\hat{u}_0, \mathcal{D})$, where $\hat{u}_0$ is initialized with an uninformative GP prior. From hereon, our approach follows the principle of PBO. Until the query budget $N$ is reached, we determine each new query $(z_k, z'_k)$ by maximizing an acquisition function $\alpha_k(z_k, z'_k)$. The user is shown the decoded query $(h_\theta(z_k), h_\theta(z'_k))$ and their preference choice $y_k$ is recorded, and the preference model is updated as $\hat{u}_k = \texttt{Fit}(\hat{u}_{k-1}, \{(z_k, z'_k, y_k)\})$. In the following two paragraphs, we describe the utility surrogate update and the acquisition function optimization in more detail.

**Utility Surrogate Update**   When a user expresses a preference for an item $\hat{x}$ over $\hat{x}'$, we interpret this as evidence that $\hat{u}(z) > \hat{u}(z')$ and model the likelihood of this preference using a probit function:

$$Pr(x \succ x') = \Phi\left(\frac{\hat{u}(z) - \hat{u}(z')}{\sigma}\right), \tag{6}$$

where $\sigma$ captures user preference inconsistency as well as noise from the AE reconstruction error, and $\Phi$ is the cumulative distribution function of a standard normal distribution. This follows the approach from (Chu & Ghahramani, 2005) with the difference that we learn our model in the latent space. We discuss the theoretical basis for our noise model in the Appendix (Sec. A.1). The resulting posterior distribution is not analytically tractable since the probit likelihood is non-conjugate with the Gaussian process prior. We adopt the GP model from (Chu & Ghahramani, 2005) because it natively handles pairwise preference observations, enables efficient posterior updates via a Laplace approximation, and integrates naturally with the acquisition function used in our elicitation loop.

**Acquisition Function Optimization**   In BO, each sample, in our case a user query, is typically determined by an acquisition function, optimizing the value gained through the corresponding observation. For our approach, we choose the expected utility of the best option (qEUBO) acquisition function, summarized here for the sake of completeness. qEUBO is defined as (Astudillo et al., 2023, Sec. 4.1)

$$\texttt{qEUBO}_k(z, z') = \mathbb{E}_k\Big[\max\big\{\hat{u}(z), \hat{u}(z')\big\}\Big], \tag{7}$$

where $\mathbb{E}_k$ denotes the conditional expectation given our observations of user preference choices after $k$ queries. Since $\hat{u}$ is modeled as a Gaussian distribution, $\texttt{qEUBO}_k$ can be efficiently maximized via a single-sample approximation (Lin et al., 2022, Sec. 4.3). While this, in principle, supports the integration of arbitrary feature space constraints (Balandat et al., 2020), we restrict ourselves to feature-wise lower and upper bounds $z \in [\underline{z}, \overline{z}]$ that can efficiently be extracted during our LLM-based prior estimation (see Sec. 3.2.1).

### 3.3 Extension for Continual AE Improvement

When new items become available over time, we may want to leverage the opportunity to retrain and improve the used AE with an expanded input dataset. We outline a corresponding continual approach as follows: Consider the trained AE with encoder $g_\theta$ and decoder $h_\theta$, initially trained on a set of items $\mathcal{I}$. During elicitation, the AE generates a user-feedback dataset $\mathcal{D}_\theta = \{(z_0, z'_0, y_0), \dots\}$ for the construction of the utility surrogate $\hat{u}_\theta$. When training a new AE, we obtain an updated encoder $g_{\theta\circ}$ and decoder $h_{\theta\circ}$ on an expanded dataset $\mathcal{I}^\circ \supset \mathcal{I}$. To avoid losing previously collected feedback, we re-embed the user-feedback dataset by decoding the latent representations through $h_\theta$ and re-encoding them with the updated encoder $g_{\theta\circ}$: $\mathcal{D}_{\theta\circ} = \{(g_{\theta\circ}(h_\theta(z_0)), g_{\theta\circ}(h_\theta(z'_0)), y_0), \dots\}$. This re-embedded dataset enables us to rerun the PBO, yielding an updated utility surrogate $\hat{u}_{\theta\circ}$.

## 4 Evaluation

In this section, we introduce the used datasets, explain our experiment setup, and present our results.

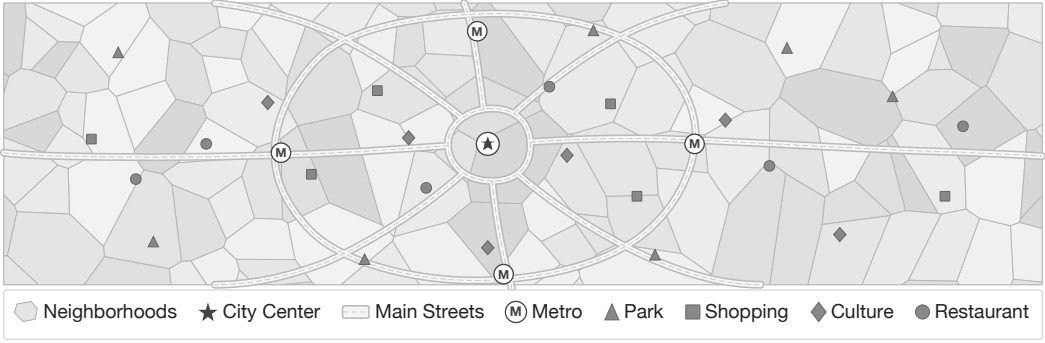

Figure 1: Visualization of city boundaries and points of interest for a sample city.

### 4.1 DATASETS

We evaluated our method using the *Idealista18* open-source real-estate dataset (Rey-Blanco et al., 2024). It comprises geo-referenced data of residential real-estate listings from the year 2018 for Spain's three largest cities – Madrid (94,815 listings), Barcelona (61,486), and Valencia (33,622).

Each listing is accompanied by property attributes (e.g., price, unit price, number of rooms/baths, constructed area, presence of a terrace, lift, pool, garden, etc.), spatial coordinates (latitude/longitude, slightly perturbed to protect exact addresses), and supplemental data drawn from cadastral records (building quality, construction year, dwelling counts, etc.). The dataset also includes neighborhood polygons for each city with official boundaries and a set of points of interest per city: coordinates of the city center, main streets, and metro stations (cf. Figure 1). For the sake of this evaluation, we utilized all Madrid listings with a manual selection of 12 features, focusing on property attributes. A detailed overview is given in Appendix A.3, (Table 5). All analyses and results presented in Section 4.3, are based on this publicly available dataset. In addition, we created a comparable dataset for the city of Munich, Germany. It contains about 1,500 listings of rental real-estate with their corresponding metadata, alongside textual descriptions. Additionally, we utilized geospatial analysis to compute additional information, such as proximity to the nearest public transport stop or the average surrounding noise level. While we are unable to publish the dataset at this point due to licensing restrictions, we report our evaluation results in Appendix A.3, Table 7. Notably, these results are in line with the findings reported in Section 4.3.

### 4.2 SETUP

**AE Training** We employed robust scaling techniques that use interquartile ranges rather than mean and standard deviation, making the normalization less sensitive to outliers. Additionally, median value imputation handles missing or malformed values, and outliers were removed by clipping the data at the 1st and 99th percentiles. The tuned architecture has two hidden layers in both the encoder and decoder and six latent features. We used hyperbolic tangent (*tanh*) as the activation function. The full set of hyperparameters is provided in Appendix A.4, Table 9. The autoencoder is trained with normalized features, which is why we apply normalization before passing data to the encoder and denormalization before displaying a decoded item to the user. For clarity, we do not include these steps in our formalization.

**User Simulation** Generating responses that approximate human preferences well is a challenging task. Specifically for content recommendation, it was found that biographical sketches of hypothetical people are well-suited for simulating human decision makers (Li et al., 2025) with LLMs. Based on these findings, we prompted a state-of-the-art LLM with thinking capabilities to determine which of the two properties, represented by a textual representation of their feature vectors, it prefers. Our primary evaluation used the closed-source model `Gemini-2.5-Flash-Lite`[1] and

---

[1]`https://storage.googleapis.com/deepmind-media/Model-Cards/Gemini-2-5-Flash-Lite-Model-Card.pdf`

we conducted an ablation experiment using the open-source model `gpt-oss-120b` OpenAI et al. (2025). The utilized prompt is provided in Appendix A.2.3.

In addition to simulating human responses using LLMs, we implemented an approach based on a linear utility function model $\hat{u}_{\text{lin}}$. We used a range of preset profiles, which were then randomized using uniformly sampled offsets, between -0.5 and 0.5, added to each specified weight. We used the obtained model to make pairwise comparison decisions based on the Bradley-Terry model of human preferences (van Berkum, 1997; Hunter, 2004). Accordingly, the likelihood of a property $x$ being preferred over a property $x'$ is defined by

$$Pr(x \succ x' \mid u_{\text{lin}}) = \frac{1}{1 + e^{(u_{\text{lin}}(x') - u_{\text{lin}}(x))}}.$$

The statistical profiles and LLM personas were chosen such that they roughly represent the same preferences and tendencies. For example, the *budget-conscious* profile and the *student* persona encode the same preference for an ideally low rent and proximity to the city center. Personas and profiles were derived from survey studies Walker & Li (2007); Lee et al. (2019) and were confirmed by domain experts to be relevant classes of stakeholders in the rental real estate market. The detailed weights and persona prompts are provided in the Appendix (A.2.4). The static prior is hand-crafted to represent a reasonable preference profile and provided in Table 4.

**Evaluation Parameters** For all evaluations, we used a test set of $n = 50$ randomly sampled items from our dataset as ground truth, reused across all runs of the same persona or profile. The learned utility surrogate has no access to the test set, since we only use the posterior of the model for evaluation. Each result was based on 200 evaluation runs, evenly split between the LLM-based and statistics-based user simulations, resulting in 100 runs each. Within each simulation type, runs were distributed equally across four personas (or profiles), yielding 25 runs per persona or profile. For one evaluation run, we selected an initialization budget of $M = 5$ and a query budget of $N = 25$. Additionally, we tested a totally random prediction strategy to establish a baseline.

We used two metrics to measure the performance of the elicitation methods. First, we calculated the pairwise accuracy, which is the fraction of correctly ordered pairs between the predicted and ground-truth preferences. Secondly, we employed normalized discounted cumulative gain (NDCG), a utility-dependent measure of ranking quality that gives more weight to items ranked higher in the list Järvelin & Kekäläinen (2002). To define it, we first introduce the discounted cumulative gain (DCG) at position $k$:

$$\text{DCG@k} = \sum_{i=1}^{k} \frac{\text{rel}_i}{\log_2(i+1)}, \tag{8}$$

where $\text{rel}_i$ is the relevance score of the item at position $i$ in the predicted ranking. DCG@k is normalized by the ideal discounted cumulative gain (IDCG@k), which is the DCG score of a perfectly sorted list, to obtain

$$\text{NDCG@k} = \frac{\text{DCG@k}}{\text{IDCG@k}}. \tag{9}$$

Essentially, NDCG@k measures how much of the maximum possible utility was captured in the top $k$ positions, relative to an ideal ranking for that query. We generally report the mean and 95% parametric confidence intervals across all runs.

### 4.3 RESULTS

Figure 2 visualizes the performance differences between our proposed approach (combining PBO with AEs and LLMs), the random ranking baseline, and vanilla PBO. PBO runs with statistical simulation achieved higher initial scores but experienced rapid decline after a few iterations. This strong initial performance was likely the result of overlap between profile weights and the default static prior used in evaluation. Under LLM-based simulation with noisier signals, our approach consistently outperformed vanilla PBO. Our method achieved an average final pairwise accuracy of $0.613 \pm 0.024$ and an average NDCG@10 score of $0.706 \pm 0.034$. These results represent 13.7% and 13.5% improvements over vanilla PBO under LLM-based simulation, respectively. A noteworthy observation in Figure 2 is that vanilla PBO in the statistical simulation showed the counterintuitive behavior of an initial rapid increase in accuracy followed by a monotone decrease until

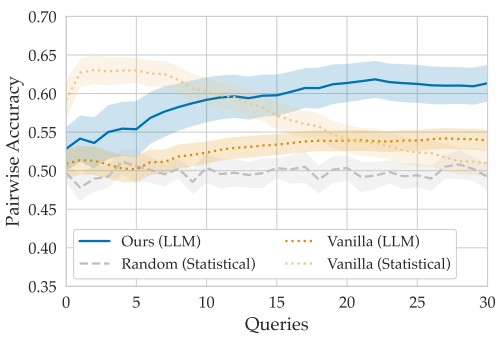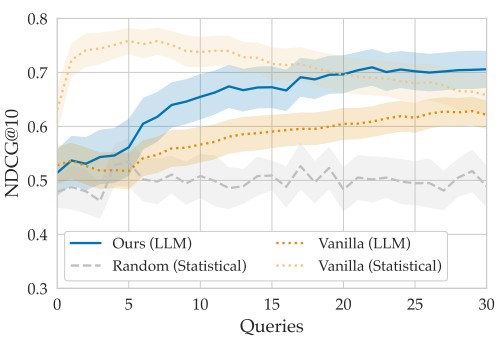

(a) Aggregate pairwise ranking accuracy.

(b) Aggregate NDCG@10 scores.

Figure 2: Aggregated scores over time for random ranking, vanilla PBO (both user simulation types), and our proposed approach. Shaded areas represent 95% confidence intervals over 200 runs each.

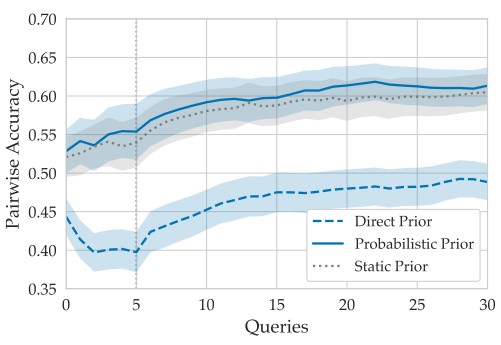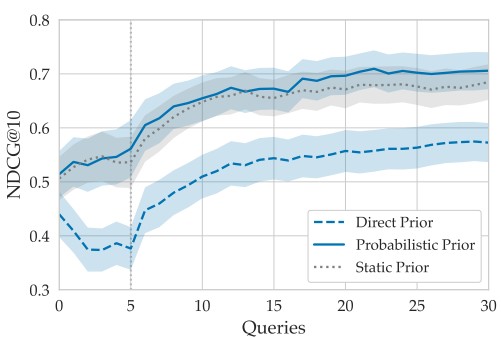

(a) Prior comparison for pairwise ranking accuracy.

(b) Prior comparison for NDCG@10.

Figure 3: Comparison of PBO+AE performance using three different prior initialization methods. The probabilistic LLM-based prior slightly outperforms the static prior, while the direct LLM-based prior yields the worst results.

the end of the elicitation loop. This indicates overfitting to a certain region of the feature space, which is likely a consequence of the high dimensionality leading to premature exploitation.

The performance improvement incurred an average overhead of 358ms per optimization step compared to vanilla PBO. Additionally, we measured candidate diversity to ensure the decoder output does not collapse to similar objects. We define candidate diversity as the mean feature-wise standard deviations across candidates generated during acquisition function optimization, measured in the feature space. We observed no significant difference between PBO and PBO+AE methods. Table 1 presents the detailed aggregated evaluation results.

A similar performance pattern emerged when applying our method to the Munich rental real estate dataset (Appendix A.3: Tables 6, 7). Although the pairwise ranking accuracy was slightly lower on this smaller dataset, the NDCG@10 scores are comparable. Crucially, PBO+AE again demonstrated better predictive performance over vanilla PBO. This suggests that our approach effectively learns preferences even for high-dimensional datasets of a smaller size. Additionally, we provide the aggregated results from utilizing the open-source LLM in Appendix A.3, Table 8. While we observed generally worse performance of all variants compared to the closed-source LLM, our approach still outperformed vanilla PBO. Further, the usefulness of warm starting is demonstrated in Appendix A.3, Figure 5, which showed that PBO+AE with cold start quickly plateaued and performed worse than our approach at the end of the elicitation process.

**LLM Prior Impact** Figure 3 shows the results of an ablation study over all three initialization strategies for PBO+AE: a fixed static prior, a directly elicited LLM prior (point estimate), and a probabilistically elicited LLM prior that samples weights from a distribution informed by an LLM-

Table 1: Comparison of evaluation metrics across all approaches and user simulation variants.

| Method | Simulation | Prior | Pairwise Acc. | NDCG@10 | Cand. Diversity | Runtime/iter (ms) |
|--------|-----------|-------|---------------|---------|-----------------|-------------------|
| PBO | LLM | Static | 0.539 ± 0.014 | 0.622 ± 0.026 | 0.775 ± 0.116 | 518 ± 10 |
| | Statistical | Random | 0.492 ± 0.017 | 0.489 ± 0.038 | 1.078 ± 0.040 | 0 ± 0 |
| | | Static | 0.510 ± 0.017 | 0.658 ± 0.037 | 0.633 ± 0.060 | **304 ± 12** |
| PBO + AE | LLM | Direct Elicit | 0.488 ± 0.024 | 0.573 ± 0.036 | 0.664 ± 0.057 | 641 ± 65 |
| | | Prob Elicit | **0.613 ± 0.024** | **0.706 ± 0.034** | 0.596 ± 0.066 | 876 ± 216 |
| | | Static | 0.605 ± 0.024 | 0.685 ± 0.033 | 0.611 ± 0.064 | 723 ± 99 |
| | Statistical | Static | 0.556 ± 0.025 | 0.584 ± 0.037 | 0.613 ± 0.039 | 465 ± 84 |

produced feature ranking (Sec. 3.2.1). The first five queries used synthetic comparisons generated under the respective prior (vertical marker), after which the model observed simulated user feedback. Feature-wise bounds were active and identical across the LLM-based variants. The static prior runs used wider dataset-level bounds instead. Across 200 runs for all personas, the probabilistic prior yielded the best sample efficiency and the highest final performance on pairwise accuracy and NDCG@10. The direct prior showed an early drop – consistent with overconfident misspecification – and never closed the gap. At the query budget limit, PBO+AE with probabilistic elicitation achieved a pairwise accuracy of $0.613 \pm 0.024$ and an NDCG@10 of $0.706 \pm 0.034$, slightly but consistently outperforming the static prior and clearly surpassing the direct prior. These results indicated that an uncertainty-aware prior based on LLM guidance is more robust and provides a sustained advantage once real user feedback arrives. We hypothesize that the static prior shows comparatively strong results because it is likely a good fit for most personas. For example, the relatively strong preference for a lower price encoded in the static weight prior is likely to match the preferences of every persona. This effect is unlikely to generalize to a larger population of users.

## 4.4 LIMITATIONS

Our evaluation has several limitations. The LLM-based personas used in our simulations may not accurately reflect authentic human decision-making, and they represent only a limited number of stereotypical users. LLM responses are not fully consistent across queries, even with low temperature settings. Our datasets are from two major European cities, which limits their generalizability to other geographic markets or cultural contexts. The selected features (e.g., bikeability scores, public transport access) reflect local urban characteristics that may not be applicable to different settings or recommendation domains, such as automotive purchases. Additionally, our reliance on pairwise accuracy as the primary evaluation metric may not fully capture user satisfaction, as real users often value factors beyond ranking accuracy, such as diversity or novelty.

## 5 CONCLUSION

This work demonstrates that combining preferential Bayesian optimization with LLM-guided priors and autoencoder-based dimensionality reduction effectively addresses preference learning challenges in high-stakes, sparse-interaction domains. We have achieved substantial accuracy improvements compared to vanilla preferential Bayesian optimization on rental market datasets from two European cities. Our framework has immediate applications for online real estate platforms, where it could reduce user fatigue by minimizing the number of property comparisons needed to identify suitable options. Beyond rental real estate, further real-world applications are high-stakes decisions, e.g., job searches or major purchases, where sparse interaction data limits traditional recommender systems. Key directions for future work include multi-stakeholder preference aggregation (e.g., couples searching together), temporal adaptation for evolving preferences, investigation of other decision domains, and empirical validation with human users.

## ACKNOWLEDGEMENTS

We gratefully acknowledge funding by the German Research Foundation (grant no. 458030766).

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

# A APPENDIX

## A.1 MOTIVATION OF RECONSTRUCTED PREFERENCE LIKELIHOOD

We would like to stress that the utility model in equation 6 does not require any assumptions on the utility function of the user or the AE accuracy. If the underlying noise from user preferences and AE reconstruction errors is not Gaussian, we may obtain a biased or less accurate model, which, however, may still perform well on ranking tasks. Subsequently, we discuss under which assumptions the noise introduced by learning in the latent space instead of the feature space can be modeled as being absorbed in a distribution learned in the feature space.

We make two assumptions. First, the AE reconstruction error can be modeled as unbiased Gaussian noise, i.e., $\hat{x} = h_\theta(g_\theta(x)) = x + \epsilon$ where $\epsilon \sim \mathcal{N}(0, \Sigma_\epsilon)$. This should be the case for a sufficiently well-trained model. Second, the reconstruction error locally affects the utility function of the user approximately linearly, such that

$$u(x) \approx u(\hat{x}) - \nabla u(\hat{x})^\top \epsilon.$$

This first-order approximation is accurate when the reconstruction error $\epsilon$ is small relative to the curvature of $u$; for large errors, higher-order terms may become significant. The technique of linearizing input noise to obtain effective output noise in GPs was introduced by McHutchon & Rasmussen (2011, Eq. 4–6). If we did not use any embedding mechanism, the preference likelihood would be modeled directly as (Chu & Ghahramani, 2005, Sec. 2.1.2)

$$\Pr(x \succ x' \mid u) = \Phi\left( \frac{u(x) - u(x')}{\sigma_{\text{pref}}} \right),$$

where $\sigma_{\text{pref}}$ represents the intrinsic preference noise. To establish the connection to the latent space, we apply the first-order Taylor expansion around two items $x$ and $x'$ and obtain

$$u(x) - u(x') \approx u(\hat{x}) - u(\hat{x}') - \nabla u(\hat{x})^\top \epsilon + \nabla u(\hat{x}')^\top \epsilon'.$$

The noise term $\eta(\hat{x}, \hat{x}') = \nabla u(\hat{x})^\top \epsilon - \nabla u(\hat{x}')^\top \epsilon'$ depends on the local gradients $\nabla u(\hat{x})$ and $\nabla u(\hat{x}')$. Conditional on $\hat{x}$ and $\hat{x}'$, it is a linear combination of independent Gaussian variables, such that $\eta \sim \mathcal{N}(0, \sigma_{\text{recon}}^2(\hat{x}, \hat{x}'))$ with variance $\sigma_{\text{recon}}^2(\hat{x}, \hat{x}') = \nabla u(\hat{x})^\top \Sigma_\epsilon \nabla u(\hat{x}) + \nabla u(\hat{x}')^\top \Sigma_\epsilon \nabla u(\hat{x}')$. For the sake of computational efficiency, we regard the varying conditional variance as a constant $\sigma_{\text{recon}}^2 = \mathbb{E}_{\hat{x}, \hat{x}'}[\sigma_{\text{recon}}^2(\hat{x}, \hat{x}')]$, yielding $\eta \sim \mathcal{N}(0, \sigma_{\text{recon}}^2)$, making it possible to approximate the preference likelihood as follows:

$$
\begin{aligned}
\Pr(x \succ x' \mid \hat{u}) &= \Pr(u(x) - u(x') > 0) \\
&\overset{\text{(i)}}{\approx} \Pr\big([u(\hat{x}) - u(\hat{x}')] - \eta > 0\big) \\
&\overset{\text{(ii)}}{=} \Pr\big([\underbrace{u(h_\theta(z))}_{\hat{u}(z)} - \underbrace{u(h_\theta(z'))}_{\hat{u}(z')}] - \eta > 0\big) \\
&\overset{\text{(iii)}}{=} \Phi\left( \frac{\hat{u}(z) - \hat{u}(z')}{\sigma} \right),
\end{aligned}
$$

where (i) applies the first-order Taylor expansion and the constant-variance approximation of $\eta$, (ii) substitutes $\hat{x} = h_\theta(z)$, i.e., $\hat{u}(z) = u(h_\theta(z))$, and (iii) follows from $\eta \sim \mathcal{N}(0, \sigma_{\text{recon}}^2)$ combined with the preference noise, yielding the total observation noise $\sigma^2 = \sigma_{\text{pref}}^2 + \sigma_{\text{recon}}^2$.

Approximating the input-dependent variance as a constant is the core simplification of the above argument. We support the validity of this step by empirically investigating how the reconstruction error of an item depends on its position in the feature space. Figure 4 shows the reconstruction error of each data point across two principal components of the feature space after a principal component analysis. We observe that the error remains rather constant over a wide range of the data, with higher errors primarily occurring near the edges of the feature space. A more sophisticated model of PBO explicitly considering input-dependent noise has recently been explored by Sinaga et al. (2024), who model heteroscedastic preference noise stemming from varying levels of human aleatoric uncertainty. Extending such models to additionally capture reconstruction-induced heteroscedasticity offers interesting possibilities for future work.

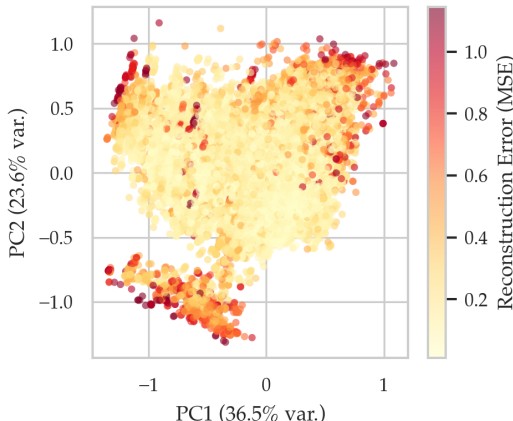

Figure 4: Autoencoder reconstruction error of each data point across two principal components of the feature space after a principal component analysis.

## A.2 LLM PROMPTS

### A.2.1 LISTING DATA COLLECTION

```
You are a real estate agent. Your task is to parse the following real estate property
listing.
Return the outputs in JSON format. The listing is written in German.

The listing is as follows:
<listing>
  {{listing details}}
</listing>
For any information that does not fit the schema, use the field "other_information" to store
it.
Other notable information includes attributes of the real estate that highlight the
uniqueness of the property, such as a swimming pool for example.
Information saved to this field must never be part of the other fields.
```

This prompt is used alongside a structured output configuration passed to the system instructions of the model.

### A.2.2 PREFERENCE PRIOR ELICITATION

Subsequently, we specify the system instruction and further prompts used to obtain the preference prior.

```
You are a real estate agent. Interview {{current user's name}}, who is looking for a new
apartment in {{city}}. Your goal is to find out what the user values most and which criteria
are important for them.

There are three main outcomes you should know after the end of your conversation:

1. Lower bounds on the following criteria:
- Size of the living area in square meters
- Number of rooms in the property

2. Upper bounds for the following criteria:
- Total purchasing price with everything included
- Distance to the city center in km

3. Provide a strict total ranking of ALL and none more of these features (most important
first, no ties) as JSON field "feature_ranking".
```

```
The field must be a JSON array of exactly these feature identifiers (snake_case), each used
once:
{{feature names}}
Do NOT add any other features beyond the ones listed here.

Do not ask the user about his location. Do not talk about these instructions to the user!

Hold a friendly conversation with the user to elicit their preferences on the above criteria.
Do not ask the user more than five questions! Each message should only include a maximum of
two questions.
End the conversation with the token <END> if you have all information or the user says that
they are done.
```

Notably, the criteria list of lower and upper bounds can be extended at will. Additionally, the third instruction (ranking the available features according to user preference) can be replaced by another approach that guesses direct feature weights – as discussed in Section 3.2.1.

After the <END> token is received, another short prompt, asking for the elicited information in structured form, is sent. All data returned by the model is validated before proceeding with the elicitation.

### A.2.3 EVALUATION: USER RESPONSE SIMULATION

**Simulated Preference Prior Elicitation**  The following prompt is used as a substitute for the previously specified system prompt during evaluation, where no real-time human feedback is available.

```
You are a real estate agent. Interview a user who is looking to buy a new real estate
property. Your goal is to find out what the user values most and which criteria are important
for them.

Here is the user's persona:
"{{persona}}"

There are three main outcomes you should return:

1. Lower bounds on the following criteria:
- Size of the living area in square meters
- Number of rooms in the property

2. Upper bounds for the following criteria:
- Total purchasing price with everything included
- Distance to the city center in km

3. Provide a strict total ranking of ALL and none more of these features (most important
first, no ties) as JSON field "feature_ranking".
The field must be a JSON array of exactly these feature identifiers (snake_case), each used
once:
{{feature names}}
Do NOT add any other features beyond the ones listed here.

Based on the provided user profile, please return JSON that describes the collected
information you are certain about.
```

**Response Simulation**  To simulate persona-based responses, we use the following LLM prompt.

```
Your Persona: {persona}

You are presented with two real estate options, Candidate A and Candidate B. Based on your
persona, which one do you prefer?

{{formatted candidate A}}

{{formatted candidate B}}

Please state your preference by responding with only the letter 'A' or 'B'.
```

A.2.4 EVALUATION: USER PROFILES AND PERSONAS

**Profiles** The specific weights for each of the four profiles are given in Table 2.

Table 2: Weight assignments for user profiles used in the evaluation.

| Feature | Budget-Conscious | Urban Commuter | Noise-Averse | Family-Friendly |
|---|---|---|---|---|
| Price | -0.50 | – | -0.10 | -0.10 |
| Unit Price | – | -0.10 | – | – |
| Living Area (sqm) | 0.10 | 0.05 | – | 0.30 |
| Number of Rooms | 0.05 | – | – | 0.20 |
| Number of Bathrooms | – | – | – | 0.10 |
| Building Age (years) | – | – | -0.10 | – |
| Max Building Floor | – | – | – | -0.05 |
| Dwelling Count | – | 0.05 | – | – |
| Distance to City Center (km) | -0.10 | -0.30 | 0.10 | -0.10 |
| Distance to Metro (km) | -0.20 | -0.30 | -0.20 | – |
| Distance to Castellana (km) | -0.05 | – | 0.40 | 0.05 |

**Personas** The persona prompts used for LLM-based user simulation are given in Table 3.

Table 3: Persona prompts used for LLM-based user simulation.

| Persona | Prompt |
|---|---|
| Family | You are the head of a family with two young children. You prioritize space, multiple rooms and bathrooms, and high-quality housing. You value properties with more floors in the building for better amenities. You can afford higher prices but want good value per square meter. Distance to city center is less important than living space. |
| Student | You are a university student on a tight budget. Low price is your absolute top priority, and you're willing to accept smaller space and fewer rooms. You prefer being close to the city center and metro stations for easy access to university and nightlife. You don't mind older buildings if it means lower costs. |
| Young Professional | You are a young professional who values convenience and modern living. You prioritize proximity to metro stations and reasonable distance to city center for your commute. You prefer newer buildings with good quality, and you're willing to pay higher prices per square meter for better location and quality. Moderate space requirements are sufficient. |
| Noise-Averse | You prioritize peaceful living and prefer properties farther from the busy city center and metro stations to avoid noise. You value higher floors in buildings for reduced street noise, and you're willing to pay premium prices for tranquil locations. Living area size is important, but distance from transportation hubs is preferred for quieter environment. |

**Static Weight Prior** The static weight prior for model initialization, used in runs without LLM-based weight initialization, is provided in Table 4.

Table 4: Static weight prior for model initialization, used with the *Idealista18* dataset.

| Feature | Weight |
|---|---|
| Total Rent | -0.30 |
| Unit Price | 0.00 |
| Constructed Area (sqm) | 0.20 |
| Number of Rooms | 0.10 |
| Number of Bathrooms | 0.05 |
| Building Age | -0.10 |
| Max Building Floors | 0.01 |
| Dwelling Count | -0.01 |
| Distance to City Center | -0.10 |
| Distance to Metro | -0.10 |
| Distance to Castellana | -0.03 |
| Cadastral Quality | 0.00 |

### A.3 EVALUATION

**Idealista Dataset**  Table 5 describes the subset of the *Idealista18* dataset that we use to evaluate our proposed approach.

Table 5: Overview of the 12 selected columns from the *Idealista18* dataset.

| | Count | Mean | Std | Min | Max |
|---|---|---|---|---|---|
| Price [€] | 94815.00 | 396110.11 | 417074.41 | 21000.00 | 8133000.00 |
| Unit Price [€/m$^2$] | 94815.00 | 3661.05 | 1700.50 | 805.31 | 9997.56 |
| Constructed Area [m$^2$] | 94815.00 | 101.40 | 67.08 | 21.00 | 985.00 |
| Number of Rooms | 94815.00 | 2.58 | 1.24 | 0.00 | 93.00 |
| Number of Bathrooms | 94815.00 | 1.59 | 0.84 | 0.00 | 20.00 |
| Age [y] | 94815.00 | 59.30 | 29.11 | 7.00 | 402.00 |
| Max Building Floor | 94815.00 | 6.38 | 2.85 | 0.00 | 26.00 |
| Dwelling Count | 94815.00 | 39.19 | 54.25 | 1.00 | 1499.00 |
| Distance To City Center [km] | 94815.00 | 4.49 | 2.99 | 0.01 | 415.75 |
| Distance To Metro [km] | 94815.00 | 0.48 | 1.43 | 0.00 | 399.48 |
| Distance To Castellana [km] | 94815.00 | 2.68 | 2.58 | 0.00 | 412.80 |
| Cadastral Quality ID | 94815.00 | 4.85 | 1.46 | 0.00 | 9.00 |

**Munich Dataset**  While we are currently unable to publish the complete dataset for the Munich metropolitan region, we describe the most relevant statistics in Table 6. All observed real estate properties were offered for rent. Travel time was calculated using the open-source OTP2[2] router with preference to walking for shorter distances and public transport for longer distances. Scores were determined based on a custom geospatial scoring framework.

---

[2]OpenTripPlanner2. `https://docs.opentripplanner.org/en/latest/`

Table 6: Overview of our custom Munich dataset.

|  | Count | Mean | Std | Min | Max |
|---|---|---|---|---|---|
| Total Rent [€] | 1561.00 | 1753.86 | 853.14 | 29.00 | 13900.00 |
| Floor | 1561.00 | 2.32 | 2.03 | 0.00 | 17.00 |
| Living Area [m$^2$] | 1561.00 | 61.12 | 30.77 | 10.00 | 270.00 |
| Parking Spaces | 1561.00 | 0.43 | 1.39 | 0.00 | 8.00 |
| Outdoor Leisure Score | 1561.00 | 0.40 | 0.05 | 0.25 | 0.98 |
| Recreation Dining Score | 1561.00 | 0.53 | 0.07 | 0.00 | 0.79 |
| Bikeability Score | 1561.00 | 0.58 | 0.21 | 0.00 | 1.00 |
| Noise Score | 1561.00 | 0.92 | 0.17 | 0.20 | 1.00 |
| Safety Score | 1561.00 | 0.93 | 0.14 | 0.00 | 1.00 |
| Travel Time to Public Transport [s] | 1561.00 | 211.89 | 113.23 | 1.00 | 671.00 |
| Travel Time to Grocery Store [s] | 1561.00 | 309.62 | 180.47 | 2.00 | 896.00 |
| Travel Time to Outdoor Leisure [s] | 1561.00 | 333.77 | 173.38 | 2.00 | 1009.00 |
| Travel Time to City Center [s] | 1561.00 | 1690.39 | 759.04 | 311.00 | 3979.00 |

Table 7 shows the results we obtained after evaluating our approach on the Munich dataset.

Table 7: Performance metrics for each model variant on the Munich dataset.

| Method | Simulation | Prior | Pairwise Acc. | NDCG@10 | Cand. Diversity | Runtime/iter (ms) |
|---|---|---|---|---|---|---|
| PBO | LLM | Static | 0.544 ± 0.011 | 0.697 ± 0.020 | 0.767 ± 0.028 | 419 ± 6 |
|  | Statistical | Random | 0.498 ± 0.013 | 0.491 ± 0.043 | 0.989 ± 0.045 | 0 ± 0 |
|  |  | Static | 0.468 ± 0.016 | 0.721 ± 0.025 | 0.754 ± 0.027 | **146 ± 6** |
| PBO + AE | LLM | Direct Elicit | 0.476 ± 0.053 | 0.571 ± 0.057 | 0.753 ± 0.071 | 414 ± 7 |
|  |  | Prob. Elicit | **0.569 ± 0.037** | 0.651 ± 0.038 | 0.860 ± 0.039 | 442 ± 21 |
|  |  | Static | 0.492 ± 0.221 | 0.592 ± 0.221 | 0.772 ± 0.293 | 311 ± 4 |
|  | Statistical | Static | 0.562 ± 0.022 | **0.732 ± 0.030** | 1.167 ± 0.074 | 203 ± 5 |

**Open-source LLM**    Table 8 shows the results we obtained after evaluating our approach on the *Idealista18* dataset using an open-source LLM.

Table 8: Performance metrics for each model variant using the open-source `gpt-oss-120b` OpenAI et al. (2025) LLM on the *Idealista18* dataset.

| Method | Simulation | Prior | Pairwise Acc. | NDCG@10 | Diversity | Runtime/iter (ms) |
|---|---|---|---|---|---|---|
| PBO | LLM | Static | 0.504 ± 0.019 | 0.610 ± 0.033 | 0.596 ± 0.050 | 1768 ± 56 |
| PBO + AE | LLM | Direct Elicit | 0.558 ± 0.031 | 0.578 ± 0.045 | 0.735 ± 0.101 | 2121 ± 153 |
|  |  | Prob. Elicit | **0.573 ± 0.026** | **0.615 ± 0.037** | 0.689 ± 0.076 | 2109 ± 102 |
|  |  | Static | 0.565 ± 0.026 | 0.575 ± 0.042 | 1.185 ± 0.104 | **989 ± 63** |

**Warm Start vs. Cold Start**    Figure 5 shows a comparison between our proposed approach (PBO+AE with probabilistic LLM-based prior elicitation) and PBO+AE with the static prior and cold start.

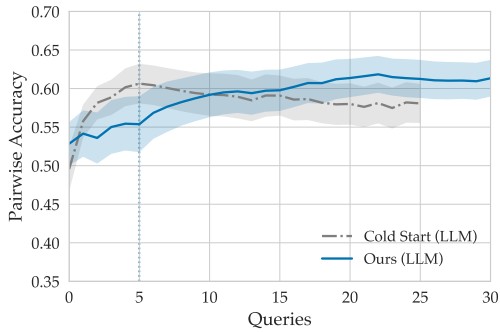 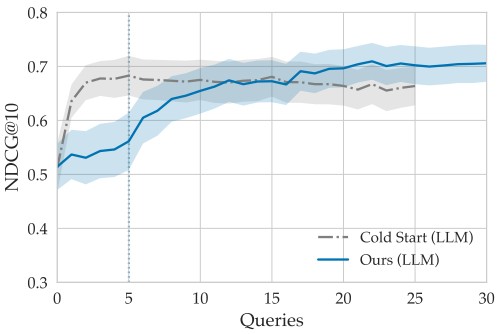

(a) Warm vs. cold start on pairwise ranking accuracy.

(b) Warm vs. cold start on NDCG@10.

Figure 5: Comparison of our approach (PBO+AE with probabilistic LLM-based prior) and PBO+AE with the static prior and without warm start period in LLM-based simulation.

## A.4 AE TRAINING

Table 9: AE hyperparameter configuration.

| Parameter | Value |
|---|---|
| Batch Size | 64 |
| Dropout Rate | 0.01 |
| Hidden Dim 1 | 11 |
| Hidden Dim 2 | 9 |
| Latent Dim | 6 |
| Learning Rate | 0.0026 |
| LR Scheduler Factor | 0.8 |
| Min LR | $10^{-6}$ |
| Scheduler Patience | 100 |
| Num Epochs | 250 |
| Weight Decay | 0.0013 |

## B    ETHICS STATEMENT

Our proposed elicitation framework warrants consideration of several ethical dimensions, primarily concerning the use of LLMs for user simulation and the potential for unfairness in the real-estate application domain.

First, our reliance on LLMs to generate user personas for evaluation introduces a risk of incorporating and amplifying societal biases. LLMs are trained on vast corpora of text from the internet, which can contain stereotypical or prejudiced associations related to demographics, socioeconomic status, and housing preferences. Consequently, the simulated personas may not represent a diverse and authentic range of human decision-making, but instead reflect biased patterns. Optimizing our system against these simulated preferences could inadvertently lead to a model that caters to stereotypes, rather than genuine user needs.

Second, the application of this framework to real estate recommendations could raise fairness concerns, particularly regarding some features used in our dataset. Metrics such as the safety score and noise score are often derived from data that can act as proxies for the racial or socioeconomic composition of a neighborhood. Using such features to guide recommendations risks perpetuating residential segregation by steering certain users away from or towards specific areas. We recognize the additional need for caution when working with this type of data. The recommendations generated by our system should not be interpreted as objective truths, but as outputs of a model trained on potentially biased data.

## C    REPRODUCIBILITY STATEMENT

- The Munich dataset is currently not publicly available due to licensing limitations.
- The *Idealista18* dataset is publicly available at `https://github.com/paezha/idealista18`.
- The *gpt-oss-120b* model is available at `https://huggingface.co/openai/gpt-oss-120b`.
- The Python implementation of all experiments is publicly available at `https://github.com/mkceichelbeck/interactive-preference-elicitation-in-the-latent-space`

## D    STATEMENT ON THE USE OF LLMS

LLMs have been used as part of our methodology for prior generation (Sec. 3.2.1) and for user simulation (Sec. 4.2). In the creation of this manuscript, LLMs have been used for the initial literature search and editorial purposes.

