# OpenReview forum: "Supporting High-Stakes Decision Making Through Interactive Preference Elicitation in the Latent Space"
_ICLR.cc/2026/Conference — ICLR 2026 Poster_

### Official Review · Reviewer_tdZs · 2025-10-29

**Soundness:** 3
**Presentation:** 4
**Contribution:** 3
**Rating:** 6
**Confidence:** 3

**Summary:**

The authors propose an interactive preference elicitation (PE) framework based on preferential Bayesian optimization (PBO).

It learns a latent utility function for the user from pairwise comparisons queried to that user.

To address the cold start issue and obtain informative probabilistic priors of feature weights for the utility function, they use an LLM-guided user interview, instead of a predefined static weight vector.

They also use an autoencoder (AE) to obtain a latent representation of lower dimension that can be used improve the sample efficiency of exploration.

They evaluate their method on rental market datasets from two European cities (Madrid, Spain and Munich, Germany).

**Strengths:**

This paper is well-motivated and addresses a real-world problem. As the authors state in the conclusion, it has immediate applications for online real estate platforms, where it could reduce user fatigue by minimizing the number of property comparisons needed to identify suitable options.

The presentation of the paper is mostly clean and easy to follow. Notation is properly introduced, and the order in which concepts are introduced is good.

The experimental results are detailed, including a high number of runs, graphs with confidence intervals, and a table with numerical metrics.

**Weaknesses:**

Experiments are limited to rental markets in two cities: Madrid, Spain and Munich, Germany.

It would be nice to see experiments beyond the rental market setting, to get a better assessment of how well the method performs in other domains.

The LLM that the authors used, Gemini 2.5 Flash-Lite, is closed-source.

The Munich dataset that the authors used is not publicly available due to licensing issues.

Minor corrections:

Page 3: Replace r << d with r \ll d.

Page 6: Replace "as activation function" with "as the activation function".

Suggestions:

Page 7: Since there's space, include the formula for NDCG@k.

**Questions:**

Page 4: "we do not explicitly denote data normalization" What does this mean?

Have you experimented with other LLMs, besides Gemini 2.5 Flash-Lite? Why not experiment with an open-source LLM? It would be good to run the experiments with other LLMs, to see how much the choice impacts performance.

How did you select the personas?

---

> ### Author Response · Authors · 2025-11-21
>
> We thank the reviewer for their thoughtful feedback and constructive suggestions.
>
> ### W1: Limited Experimental Domains (Only Rental Markets)
>
> We fully acknowledge this limitation in the manuscript and suggest investigating further domains as future work.
> We specifically chose the rental real estate market domain because it is representative of high-stakes, sparse-feedback decisions and is not well-explored in the literature.
>
> ### W2/Q2: Closed-Source LLM (Gemini 2.5 Flash-Lite)
>
> We agree that experimenting with a different LLM would give insights into the robustness of our approach.
> We will replicate our core experiments on the Madrid dataset with an SOTA open-source LLM and include the results in the revised manuscript.
>
> ### W3: Munich Dataset Not Publicly Available
>
> We acknowledge the closed-source nature of the Munich dataset as a limitation to reproducibility and are actively working to obtain permission to publish the dataset.
>
> ### Minor Corrections
>
> Thank you for these suggestions.
> We will implement them in the revised version of the manuscript.
>
> ### Q1: "We do not explicitly denote data normalization" (Page 4)
>
> Thank you for pointing out this potential for clarification. We will modify the statement in the revised manuscript:
> > The autoencoder is trained with normalized features, which is why we apply normalization
> > before passing a data point to the encoder and denormalization before displaying a decoded
> > item to the user. For clarity, we do not include these steps in our formalization.
>
> ### Q3: Persona Selection Process
>
> We appreciate this question and will include the corresponding clarification in the revised manuscript:
> >Personas and profiles are derived from survey studies (Walker & Li (2007); Lee et al. (2019)) and confirmed by domain experts to be relevant classes of stakeholders in the rental real estate market.
>
> Joan L. Walker and Jieping Li. Latent lifestyle preferences and household location decisions. Journal of Geographical Systems, 9(1):77–101, 2007.
>
> Yongsung Lee, Giovanni Circella, Patricia L. Mokhtarian, and Subhrajit Guhathakurta. Heterogeneous residential preferences among millennials and members of generation X in California: A latent-class approach. Transportation Research Part D: Transport and Environment, 76:289–304, 2019.

---

### Official Review · Reviewer_EyRm · 2025-10-30

**Soundness:** 3
**Presentation:** 2
**Contribution:** 3
**Rating:** 6
**Confidence:** 2

**Summary:**

The paper targets high-stakes, sparse-interaction decisions such as housing selection, where conventional recommender systems fail due to limited feedback and high-dimensional features. To address this, it proposes an interactive preference elicitation framework that combines Preferential Bayesian Optimization (PBO) with two key components: LLM-based probabilistic priors, which interpret natural-language interviews to initialize user utility weights and mitigate cold start; and an Autoencoder-based latent representation, which reduces dimensionality for efficient exploration. The system learns a latent utility model from pairwise user comparisons using a Gaussian-process surrogate and qEUBO acquisition for adaptive querying. Evaluations on real-estate datasets from Madrid and Munich show that latent-space PBO improves ranking accuracy and that LLM-guided priors significantly enhance sample efficiency and cold-start performance.

**Strengths:**

The paper tackles an important and realistic problem of learning preferences in high-stakes, sparse-feedback decision settings. It proposes a coherent framework that integrates LLM-based prior elicitation, Autoencoder latent-space learning, and Preferential Bayesian Optimization into a unified probabilistic approach. The method is well-founded, combining Gaussian Process modeling with qEUBO-based active querying for efficient preference learning. Experiments on real-estate datasets show clear improvements in ranking accuracy and cold-start performance, demonstrating both methodological novelty and strong practical relevance.

**Weaknesses:**

## Weaknesses
A central modeling limitation lies in the treatment of noise within the preference likelihood. The paper explicitly assumes that both the user’s preference inconsistency and the autoencoder (AE) reconstruction error can be modeled as *jointly Gaussian and additive*. In Appendix A.1 (lines 593 page 11), the authors state that the decoder output can be written as $ \hat{x} = h_\theta(g_\theta(x)) = x + \epsilon $ with $ \epsilon \sim \mathcal{N}(0, \Sigma_\epsilon) $, and that the user’s utility varies locally linearly, $ u(\hat{x}) \approx u(x) + \nabla u(x)^\top \epsilon $. Combining these yields a single Gaussian error term $ \eta $ with variance $ \sigma^2 = \sigma_{\text{pref}}^2 + \sigma_{\text{recon}}^2 $ in the Probit likelihood, expressed as
$ P(z \succ z') = \Phi\!\left(\frac{u(x) - u(x')}{\sigma}\right) $,
where $ \sigma $ captures both preference noise and AE uncertainty (page 5).
While elegant, this simplification conflates heterogeneous uncertainty sources—human inconsistency, model reconstruction bias, and latent-space distortions—into one scalar variance term. It presumes homoscedastic, isotropic noise and local linearity of the utility function, which are rarely valid in complex, nonlinear real-estate feature spaces. In practice, AE errors vary significantly across regions of the feature manifold, and user responses exhibit contextual, multimodal variability. This global Gaussian assumption therefore risks *underestimating epistemic uncertainty*, leading to overconfident posterior estimates under the Laplace approximation and potentially premature exploitation in qEUBO selection.

Beyond these statistical concerns, the model’s abstraction of user feedback neglects potential heteroscedastic and structured noise patterns. Real users exhibit region-dependent reliability—for example, being consistent on low-price listings but noisy on high-price ones—yet the framework treats all feedback as equally noisy.

**Questions:**

See Weaknesses

---

> ### Author Response · Authors · 2025-11-21
>
> We thank the reviewer for their thoughtful feedback and constructive suggestions.
>
> ### W1: Gaussian Noise Assumption and Heteroscedastic Uncertainty
>
> We agree that modeling uncertainty in a single parameter would only yield a perfect model under several assumptions that are unlikely to be fulfilled in practice. We would like to emphasize that this limitation applies to virtually all applications of Gaussian Process models and does not imply that the learned model is impractical. However, we will clarify the underlying assumptions of our model in the revised text.
>
> Our approach introduces noise from the reconstruction error of the autoencoder. The more independent the reconstruction errors are from their input items, the more accurate the used model is. In the revised paper, we will include a plot showing the reconstruction error of each data point across the two principal components of the feature space after a PCA. The plot indicates that the reconstruction error remains relatively constant over a wide range of the data, with higher errors primarily occurring near the edges of the feature space.
>
> A more sophisticated model of Preference Bayesian Optimization explicitly considering heteroscedastic noise has, to the best of our knowledge, not been formulated and presents interesting potential for future work.

---

### Official Review · Reviewer_9kWw · 2025-11-01

**Soundness:** 3
**Presentation:** 3
**Contribution:** 3
**Rating:** 6
**Confidence:** 3

**Summary:**

The paper proposes an interactive preference elicitation framework for complex, infrequent decisions like housing purchase choice. It combines LLMs (to generate personalized priors from natural language) and an autoencoder (to reduce feature dimensionality) within a PBO that learns user utility from pairwise comparisons in real time, evaluated on rental datasets from two European cities.

**Strengths:**

The paper is well-written with clear structures. The problem itself is well motivated.

**Weaknesses:**

The intuition for the selection of each structure for each purpose could be strengthened. Please refer to the questions.

**Questions:**

- In line 64-67, what is the purpose/intuition of still using LLM for the continuous feature space? Even after reading Section 3.2.1, my question about this does not go away.
- In line 82, you may want to elaborate/define the meaning of ``interactive preference elicitation'' which has appeared in the abstract & title but nowhere else before line 82 in the intro.
- What is the purpose of the warm-start, and any more empirical results to show the importance of this?
- In the experiment, what is the intuition for the vanilla (statistical) having an opposite trend (increase first and then decrease over the number of queries) with other benchmarks?

I might raise my rating after seeing the rebuttal.

---

> ### Author Response · Authors · 2025-11-21
>
> We thank the reviewer for their thoughtful feedback and constructive suggestions.
>
> ### Q1: LLM Usage for Continuous Feature Space + Importance of Warm Start
>
> In high dimensional feature spaces, it is challenging for human users to express numerical preferences. This is why previous work has employed LLMs allowing users to express their preferences in natural language. However, such a natural language representation is rather coarse and, therefore, inefficient for continuous feature spaces.
> To still leverage natural language understanding, we use the LLM-based warm start. Here, the coarseness is acceptable since warm-starting with a simplified linear utility model only aims to roughly align the preference model before the first query is given to the user. This should make the initial set of queries more relevant and improve learning progress.
>
> Our results in Table 2 show that the used prior makes a substantial difference. We are additionally working on including a baseline without any warm start.
>
> ### Q2: Definition of "Interactive Preference Elicitation"
>
> Thank you for this suggestion. In the revised version, we will introduce the term in the beginning of the introduction.
>
> ### Q3: Vanilla Statistical Prior Trend (Increase then Decrease)
>
> This is an excellent question. Our hypothesis is that this observation indicates overfitting to a certain region of the feature space. This is likely a consequence of the high dimensionality and associated issues of overconfident estimates and premature exploitation. We will add a short discussion of this observation to the revised manuscript.

---

### Meta-Review · Area_Chair_qkPM · 2026-01-06

**Summary:**

This paper develops an interactive framework for preference elicitation on top of preference Bayesian optimization by combining LLMs for a prior to avoid cold start and an autoencoder to reduce the dimensionality for explicit exploration. qEUBO was used for adaptive querying. All reviewers were initially positive and though the framework was useful and the problem was realistic. The author reply was responsive to concerns about modeling assumptions and using a closed LLM.

**Reviewer Concerns:**

9kWw - Using the LLM for continuous feature space, Purpose of warm start, Potential overfitting

EyRm - Modeling limitation in the additive noise.

tdZs - Limited experiments, closed-source LLM, one of the two datasets is not open.

**Reviewer Scores:**

I think 9kWw would raise their score and the other would remain the same.

---

### Decision · Program_Chairs · 2026-01-26

Accept (Poster)